# A Novel Framework for Road Traffic Risk Assessment with HMM-Based Prediction Model

**DOI:** 10.3390/s18124313

**Published:** 2018-12-07

**Authors:** Xunjia Zheng, Di Zhang, Hongbo Gao, Zhiguo Zhao, Heye Huang, Jianqiang Wang

**Affiliations:** 1State Key Laboratory of Automotive Safety and Energy, Tsinghua University, Beijing 100084, China; zhengxj15@mails.tsinghua.edu.cn (X.Z.); hhy18@mails.tsinghua.edu.cn (H.H.); 2School of Information Engineering, Zhengzhou University, Zhengzhou 450001, China; iedzhang@zzu.edu.cn; 3Jiangsu Key Laboratory of Traffic and Transportation Security, Huaiyin Institute of Technology, Huai’an 223001, China; zhaozg@hyit.edu.cn

**Keywords:** road traffic risk assessment, intelligent transportation system, hidden Markov model, V2X

## Abstract

Over the past decades, there has been significant research effort dedicated to the development of intelligent vehicles and V2X systems. This paper proposes a road traffic risk assessment method for road traffic accident prevention of intelligent vehicles. This method is based on HMM (Hidden Markov Model) and is applied to the prediction of steering angle status to (1) evaluate the probabilities of the steering angle in each independent interval and (2) calculate the road traffic risk in different analysis regions. According to the model, the road traffic risk is quantified and presented directly in a visual form by the time-varying risk map, to ensure the accuracy of assessment and prediction. Experiment results are presented, and the results show the effectiveness of the assessment strategies.

## 1. Introduction

Road traffic accidents are one of the largest and most serious public health problems in the world. A report published by the World Health Organization in 2015 stated that around 1.2 million people lost their lives in crashes on the roads around the world each year and was the leading cause of death among the young crowd 15 to 29 years of age. However, almost all road traffic accidents have a connection with human mistakes. Many researchers attempt to address this issue. The research [1] conducted by Daimler-Benz and National Highway Traffic Safety Administration (NHTSA) in 50% to 90% rear-end and intersection accidents can be prevented if the vehicle can recognize the risk one second in advance. In this regard, the risk assessment for road traffic is of great importance for accident avoidance. Generally, road traffic assessment has two types of typical methods in both macroscopic and microscopic levels. On the one hand, road traffic risk is usually measured according to historical accident data [2,3,4], and statistical models according to regression analysis such as Poisson, Poisson-gamma, and Poisson-Lognormal [5,6]. On the other hand, researchers proposed microscopic road risk assessment approaches by distinguishing the longitudinal and lateral of vehicle motion directions. Microscopic road traffic risk indicators in the longitudinal [7,8,9] include time to collision (TTC), inverse time to collision (TTCi), time headway (THW) threshold and the deceleration rate to avoid crash (DRAC), and those in the lateral include car’s current position (CCP), time to lane cross (TLC), and variable rumble strip (VRBS) [10,11,12]. Meanwhile, researchers have developed a number of collision avoidance methods such as dynamic window approach [13] performance metrics [14].

However, existing methods leave room for argument. The macroscopic study of road risk assessment often needs to collect the accident data by considering many categories of injury severity and crash type, which is complicated and costly. Meanwhile, such methods cannot be used for real-time risk assessment. Similarly, traffic risks cannot be defined as continuous variables using the above microscopic methods which are artificially classified into longitudinal and lateral directions.

In recent decades, artificial potential field (APF) methods are widely used in robotics. Due to its advantages in environment description, APF methods are widely used in path planning, obstacle avoidance, and road traffic risk assessment for intelligent vehicles [15,16,17]. In these studies, road traffic risk can be shown as a visible distribution by using APF’s risk map, where the value of the artificial potential energy denotes the road traffic risk. Then the larger the value is, the higher the road risk is. However, the artificial potential energy can only represent the changes and trends in the road traffic risk. If we want to evaluate the road traffic risk accurately, a large number of undetermined parameters in the model need to be calibrated [16,17]. 

Recently, the V2X technologies (including V2I, V2V, V2R) are improving with the rapid development of wireless communication technology such as the dedicated short-range wireless communication technology (DSRC) [18,19]. The subject vehicle can capture the information of around road users, infrastructures, and road conditions effortlessly based on V2X technologies during driving. It provides an opportunity for further development of road risk assessment methods.

To date, few researchers have proposed practical and straightforward methods for road traffic risk assessment. Meanwhile, there are some drawbacks to the existing road traffic risk assessment methods. The timeliness and accuracy of traffic risk assessment cannot be guaranteed simultaneously. Besides, most of the applications of these methods are limited to some simple scenarios. This article aims to improve this situation. Thus, we first present a novel road traffic risk assessment framework combined with the equivalent force [20] and HMM-based prediction algorithm. Then, we define the range of the potential traffic risk caused by vehicles according to vehicle kinematics which can describe the traffic risk consistent with human’s intuition perception of risk. Finally, the field experiment results show that the time-varying traffic risk map can illustrate the road traffic risk directly.

The rest of this paper is organized as follows. In Section 2, we describe the road traffic risk quantitatively with considering the kinetic energy of the road user and present a traffic risk range model according to the steering angle and turning-probability. In Section 3, an algorithm of road traffic risk assessment is proposed. Section 4 introduces three experimental vehicles and describes the results of the field experiments. Section 5 presents the conclusions of this study.

## 2. Proposed Methods

Traffic always flows in a predetermined direction much as free objects always fall to the ground [21]. In a car-following scenario, a collision usually occurs between the preceding vehicle of the trailing vehicle and the rear vehicle of the lead vehicle. The severity of the crash is a strong positive correlation with the velocity of two vehicles. As is well known, if a collision occurs at some point, the kinetic energy stored will be released, transferred, and lead to elastic and plastic deformation. Moreover, the kinetic energy denotes the risk from the energy of the vehicle itself, and the accident is an abnormal transfer of energy according to the energy transfer theory [22]. Therefore, in this study, we consider the kinetic energy of road users as one of the essential causations of the potential traffic risk.

### 2.1. The Road Traffic Risk Formulation of Road Users

The kinetic energy of a moving vehicle on the road is $\frac{1}{2}m_{i}v_{i}^{2}$. The vehicle will create potential traffic risks ahead according to the above consideration, and we describe the potential traffic risks as follows:(1)$$E_{i}=\frac{1}{2}m_{i}v_{i}^{2}=\frac{1}{2}m_{i}v_{i}\cdot v_{i}=\frac{1}{2}m_{i}v_{i}\cdot \frac{\left(v_{i}-0\right)}{\Delta x_{i}}\cdot \Delta x_{i}$$
where, $E_{i}$ denotes the kinetic energy of vehicle i, $m_{i}$, and $v_{i}$ are described as the mass, and velocity of vehicle i, and the $\Delta x_{i}$ is the arc length between vehicle i and other position in the traffic environment according to the steering angle.

Let $F_{i}=\frac{1}{2}m_{i}v_{i}\cdot \frac{\left(v_{i}-0\right)}{\Delta x_{i}}$, therefore,
(2)$$E_{i}=F_{i}\cdot \Delta x_{i}$$
where, $F_{i}$ represents the equivalent force caused by the vehicle i.

In a car-following scenario including vehicle i and vehicle j. Here we use $E_{ij}$ to describe this traffic risk between these two vehicles.
(3)$$E_{ij}=\frac{1}{2}m_{i}v_{i}\cdot \left(v_{i}-v_{j}\right)\cdot \frac{\left|x_{i}-x_{j}\right|}{\left|x_{i}-x_{j}\right|}=\frac{1}{2}m_{i}v_{i}\cdot \frac{v_{i}-v_{j}}{\left|x_{i}-x_{j}\right|}\cdot |x_{i}-x_{j}|$$
where, $v_{j}$ and $x_{j}$ denote the velocity and the longitudinal position of the vehicle j. We set $F_{ij}=\frac{1}{2}m_{i}v_{i}\cdot \frac{v_{i}-v_{j}}{\left|x_{i}-x_{j}\right|}$, which denotes the internal equivalent force between these two vehicles. The $\left(v_{i}-v_{j})/|x_{i}-x_{j}\right|$ indicates the relative velocity divided by the relative distance between the vehicle i and j. In addition, the traffic risk will generate if the value of $v_{i}-v_{j}$ is positive. Otherwise, there is only potential traffic risk between the two vehicles. 

Generally, we define the quotient of relative speed and the relative position between two vehicles as the inverse of time to collision (TTCi) in car-following scenarios. Therefore, the above formula can be expressed as follows: (4)$$E_{ij}=\frac{1}{2}m_{i}v_{i}\cdot TTCi\cdot |x_{i}-x_{j}|$$
similarly, we set $F_{ij}=\frac{1}{2}m_{i}v_{i}\cdot TTCi$. In addition, we consider the possibility of collisions between vehicles and define the following vehicle as an active-collision participant (ACP) and the leading vehicle as a passive-collision participant (PCP). We can use these two terms to describe the role of vehicles in a traffic environment directly.

In this study, we use the equivalent force $F_{i}$ (caused by one road user) and $F_{ij}$ (between two road users) to describe the road traffic risk. The greater the equivalent force of one road user is, the greater the risk of traffic is. Due to the regularity of vehicle motion, the traffic risk caused by vehicles has a certain range to a certain extent. We will analyze the influence range of the traffic risk in detail in the next section.

### 2.2. The Definition of Road Traffic Risk Range

Road traffic risks are often caused by the interaction between vehicles with road traffic environments. It has relation with the maneuver states of vehicles and the road environment conditions. If there is only one vehicle driving on the road, it creates potential traffic risks for the road environment, and the potential traffic risk will evolve to the traffic risk if another vehicle occurs. Then, if one of the two drivers are not aware of the traffic risk caused by the other, or if they do not control the traffic risk caused by themselves in the driving process, the traffic risk will further increase and even evolve into a traffic accident. To avoid this situation, we analyze the range of road traffic risk in this study. In the meantime, we established a mathematical model for this risk range description.

We assume that all drivers follow traffic rules, for example, when not in the right situation they do not reverse, turn and change lanes to drive forward according to the rules. Meanwhile, the driving processes are steady. We take the velocity and steering angle of the vehicle as continuous variables. Therefore, we can predict the position of the vehicle over a period, on which the expected paths can be projected as shown in Figure 1. 

Symbol F indicates the equivalent force at each position, and $R_{i}$ denotes the turning radius of the vehicle, it can be calculated according to a two-wheel vehicle model as follows:(5)$$R_{i}\left(t\right)=\left[1+K_{i}v_{i}^{2}\left(t\right)\right]\frac{L_{i}}{\delta_{i}\left(t\right)}$$
where $K_{i}$, $L_{i}$, $\delta_{i}$, and $v_{i}$ indicate the stability factor, the wheelbase, the steering angle, and the velocity of the vehicle, respectively.

When the vehicle drives at a constant velocity with negligible side slip angle, the predicted positions $\left(x_{ip},\;y_{ip}\right)$ at a time horizon $t_{p}$ with a command steering angle $\delta_{i}$ can be calculated as follows:(6)$$\left[\begin{array}{c} x_{ip}\\ y_{ip} \end{array}\right]=\left[\begin{array}{c} x_{t_{0}}+\int_{t_{0}}^{t_{p}}v_{i}\left(t\right)\cdot cos\frac{v_{i}\left(t\right)\cdot \Delta t}{R_{i}\left(t\right)}\;dt\\ y_{t_{0}}+\int_{t_{0}}^{t_{p}}v_{i}\left(t\right)\cdot sin\frac{v_{i}\left(t\right)\cdot \Delta t}{R_{i}\left(t\right)}\;dt \end{array}\right]$$
where, Δt=1 s. It indicates the unit time. 

It is assumed that vehicle i always drives stability. The motion state of the vehicle should be based on the road conditions, and it must be subject to the following functions:(7)$$\begin{array}{c} \sqrt{F_{X,i}^{2}+F_{Y,i}^{2}}=\varphi F_{Z,i}\\ F_{X,i}=m_{i}gf+\frac{C_{D,i}A_{i}\cdot v_{i}^{2}\left(t\right)}{21.15}\\ F_{Y,i}\geq m_{i}\frac{v_{i}^{2}\left(t\right)}{R_{i}\left(t\right)} \end{array}$$
where, $F_{X}$, $F_{Y}$ and $F_{Z}$ denote the longitudinal, the lateral, and the ground reaction forces of vehicle i respectively, φ the adhesion coefficient, f the rolling resistance coefficient; $C_{D,i}$ and $A_{i}$ are the air resistance coefficient and the windward area of vehicle i respectively.

According to Equation (5) and (7), the relationship between the steering angle $\delta_{i}$ and velocity $v_{i}\;$can be derivated as follows: (8)$$\begin{array}{l} \left|\delta_{i}\left(t\right)\right|\\ \leq [\frac{K_{i}L_{i}}{m_{i}}+\frac{L_{i}\;}{m_{i}\cdot v_{i}^{2}\left(t\right)}]\sqrt{\varphi^{2}F_{Z,i}^{2}-m_{i}^{2}g^{2}f^{2}-2m_{i}gf\cdot \frac{C_{D,i}A_{i}\cdot v_{i}^{2}\left(t\right)}{21.15}-{\left[\frac{C_{D,i}A_{i}\cdot v_{i}^{2}\left(t\right)}{21.15}\right]}^{2}} \end{array}$$

In addition, each vehicle has a certain steering angle limit $\delta_{max}$ according to the mechanical structure. Generally, $\delta_{max}\in \left[-\pi /4,\pi /4\right]$ for a passenger car.
(9)$$\left|\delta_{i}\left(t\right)\right|\leq \delta_{max}$$

Therefore, the possible path of the vehicle i have a certain boundary according to the steering angle interval. As presented in Figure 2, the black dashed line is the predictive trajectory corresponding to the left and right limit steering angle. When the vehicle i is driving straight on the road, the driver has three possible manipulations, including keeping straight driving, turning to the left lane or the right lane. Obviously, the potential road traffic risk caused by the vehicle can only existing in this area.

Assume that $\delta_{k}$ and $p_{k}$ indicate the steering angle and the turning-probability, respectively. The turning-probability $p_{k}$ can be explained as follows:(10)$$\overset{n}{\underset{k=-n}{\sum }}p_{k}=1$$
(11)$$\delta_{k}=k\cdot \Delta \delta ,\;k\in \left[-n,n\right]$$
where, k,n∈Z. Δδ is the increment of the steering angle. In addition, $\delta_{0}$ indicates straight driving along the road lane, $\delta_{k}$ means turning left if k is positive. Otherwises, the $\delta_{k}$ means turning right.

However, it is challenging to directly obtain the steering angle of the vehicle and attribute a corresponding value to the turning probability. Therefore, we established an HMM-based prediction model, and we will describe it in detail in the next section.

### 2.3. HMM-Based Prediction Model for Steering Intention Recognition

It is simplistic and crude if we judge the driver’s steering intention (keep straight driving, change to left/right lane) only by considering the expectation of the driver to go straight, turn left, or turn right during driving process. Moreover, this method of classification is just a qualitative description of the driver’s intention of steering behavior; it is not enough to accurately depict the steering behavior of the driver. To solve this problem, we proposed a discretization method of steering angle. Explicitly, we define the steering angle belongs to a specified interval—that is $\delta \in \left[\delta_{-n},\delta_{n}\right]$, where the $\delta_{-n}$ denotes the right limit steering angle, and $\delta_{n}$ the left limit steering angle. Meanwhile, we divide both the left and right side of the vehicle centerline equally into n portions respectively, as shown in Figure 2. Hence there are 2n parts, and they are corresponding to 2n hidden states in the hidden Markov model. When the desired steering angle is located in the kth subinterval $\left(\delta_{k-1},\delta_{k}\right]$ in the left side of the vehicle centerline, that means the corresponding driver’s turning intention is S=k,k∈[1,n]. Similarly, when the desired steering angle is located in the −kth subinterval $\left[\delta_{-k},\delta_{-k+1}\right)$ in the right side of the vehicle centerline, that means the corresponding driver’s turning intention is S=−k,k∈[1,n]. Obviously, the greater the n is, the more intensive divided the interval $\left[\delta_{-n},\delta_{n}\right]$ are; and the more precise the driver’s steering intentions are. Generally, the driver’s steering intention is mainly affected by road condition information, traffic information, weather conditions, and the driving dynamic state of the subject vehicle and other vehicles during driving process. Within these factors, the driving dynamic state of front vehicles (including the ahead vehicles in the current and adjacent lanes) has a greater impact on the driver of the subject vehicle. In this study, we describe the driver’s steering intention as a random process—that is $\left\{S_{t},t\geq 0\right\}$. According to the above discretization method, we assume the probability distribution column for this random process $\Pi =\left(p_{k}\right)$ as follows:(12)$$\Pi \left(S_{t}=k\right)=p_{k},\;k=-n,\;\cdot \cdot \cdot -1,1,\;\cdot \cdot \cdot ,n$$
where, $\sum_{-n}^{n}p_{k}=1,k\neq 0$. Then the probability of the driver’s intention to change from one state to another can be described by the state transition matrix A: (13)$$A=\left[\begin{array}{cc} \begin{array}{ccc} \delta_{n,n} & \cdots & \delta_{n,1}\\ \vdots & \ddots & \vdots \\ \delta_{1,n} & \cdots & \delta_{1,1} \end{array} & \begin{array}{ccc} \delta_{n,-1} & \cdots & \delta_{n,-n}\\ \vdots & \ddots & \vdots \\ \delta_{1,-1} & \cdots & \delta_{1,-n} \end{array}\\ \begin{array}{ccc} \delta_{-1,n} & \cdots & \delta_{-1,1}\\ \vdots & \ddots & \vdots \\ \delta_{-n,n} & \cdots & \delta_{-n,1} \end{array} & \begin{array}{ccc} \delta_{-1,-1} & \cdots & \delta_{-1,-n}\\ \vdots & \ddots & \vdots \\ \delta_{-n,-1} & \cdots & \delta_{-n,-n} \end{array} \end{array}\right]$$
where, $\delta_{i,j}=P(S_{t+1\;}=j|S_{t}=i)$ denotes that the probability of the steering angle at time t is located in the ith interval $\left[\delta_{i-1},\delta_{i}\right)$ and the steering angle at time t+1 is located in the jth interval $\left[\delta_{j-1},\delta_{j}\right)$ is equal to $\delta_{i,j}$.

Although the driver’s steering intention is difficult to observe during the driving process, it is mainly affected by the distance between the subject vehicle and the vehicle in front of the current lane and adjacent lanes. In this study, we set the time headways of front vehicles in the current lane, and adjacent lanes are selected as the observation state in the hidden Markov model. Similarly, we divide the interval of time headways of the front vehicle in the current lane—that is $[T_{e_{min}},T_{e_{max}}]$, the left adjacent lane—that is $[T_{l_{min}},T_{l_{max}}]$, and the right adjacent lane—that is $[T_{r_{min}},T_{r_{max}}]$, equally into h portions based on the aforementioned discretization method, respectively. Therefore, there are $h^{3}$ observation states $\left\{T_{em},T_{lm},T_{rm}\right\}$, where the $m\in \left[1,h^{3}\right]$. Then, we sort these observations state, and the output state is O=i when the desired observation state located in the ith state. Therefore, we use the conditional probability $t_{i,j}=P(O_{t}=j|S_{t}=i)$ to describe the relationship between the observed state and the hidden state, and define the confusion matrix B to be:(14)$$B=\left[\begin{array}{c} \begin{array}{ccc} t_{n,1} & \cdots & t_{n,m}\\ \vdots & \ddots & \vdots \\ t_{1,1} & \cdots & t_{1,m} \end{array}\\ \begin{array}{ccc} t_{-1,1} & \cdots & t_{-1,m}\\ \vdots & \ddots & \vdots \\ t_{-n,1} & \cdots & t_{-n,m} \end{array} \end{array}\right]$$

Therefore, the hidden Markov model λ=(A,B,Π) can be used to simulate the driver’s intention during driving. Accurately, it describes as follows:(15)$$S_{t+1}=AS_{t}$$
(16)$$O_{t}=BS_{t}$$
where, $S_{t}$ denotes the hidden state—that is, the steering angle $\delta_{k}$; $O_{t}$ the observation state—that is, the time headway between the subject vehicle and the front vehicles (including the ahead vehicles in the current and adjacent lanes), it is defined as $O_{t}=\left\{T_{e},T_{l},T_{r}\right\}$.

## 3. Algorithm for Road Traffic Risk Assessment

The road traffic risk is described detailly in Section 2, and the steering angle of each vehicle is predicted by the HMM algorithm in Section 2.3. We use equivalent force to describe the potential road traffic risks. Besides, the safety level of the road environment can be quantified based on the analysis of the distribution of the equivalent force. The value of equivalent force decreases with the distance between the predicted point and the target vehicle increases. Similarly, the amount of equivalent force is decremented laterally to both sides, and the weight of the equivalent force is specified as $w_{k}$ in Figure 3. At the same time, the weight $w_{k}$ are defined as follows: (17)$$w_{k}=\frac{p_{k}}{\max\left(S_{t}\right)}$$
where, k∈{−n, …−1,1,…,n} and k,n∈Z. $p_{k}$ denotes the probability of vehicle stayed at an angle $\delta_{k}$ by the driver in the next moment.

Based on Equation (17), the equivalent force in each predictive position subject to:(18)$$F_{ki}=\frac{E}{\Delta x_{i}}\cdot w_{k}=\frac{\frac{1}{2}w_{k}m_{i}v_{i}^{2}}{\Delta x_{i}}$$

Therefore, we define the road traffic risk as the magnitude of the equivalent force at any point in the traffic environment. When the equivalent force of any road user is too large, it indicates that the traffic risk is increasing. For a real road, the HMM algorithm is used to study the driving process of vehicles on this road section over a period, which can predict the motion state of each vehicle in the next period to some extent, and then calculate the road traffic risk on this road section in real time. For example, the predicted results show that the orange vehicle will change lanes, so the current distribution of road traffic risk can be expressed in the time-varying traffic risk map directly as shown in Figure 4.

The algorithm of the road traffic risk assessment is shown in Algorithm 1. Firstly, the time-varying traffic risk map is established according to our mathematical model by input the parameters. Then, we define the threshold force of the warning $F_{warning}$ and braking $F_{braking}$. Finally, the algorithm output instructions of warning or active braking to the actuator.

**Algorithm 1.** Risk warning and active braking.
**Input**
$m_{i}$, $v_{i}$, $x_{i}$, $y_{i}$, φ, f, $C_{D,i}$, $A_{i}$, λ, n
**Output**

F
1Get the dynamic vehicle states in the surrounded environment.2**for**i=1 to n (n denotes the number of road users) **do**3Calculate the kinetic energy of each moving object *i*: $E_{i}=\frac{1}{2}m_{i}v_{i}^{2}$
4Calculate the equivalent force $F_{i}$ based on $E_{i}=F_{i}\cdot \Delta x_{i}$
5Calculate the weight$\;w_{k}$ through the improved HMM algorithm6Define the road traffic risk as the magnitude of the equivalent force:$\;F_{ki}$
7Initialize the total force in the traffic environment: F=0
8**if***i* is in the car-following scenario9
$F_{ki}=\frac{E}{\Delta x_{i}}\cdot w_{k}=\frac{\frac{1}{2}w_{k}m_{i}v_{i}^{2}}{\Delta x_{i}}$
10Calculate $F=F+F_{ki}$ (the for range only considers the same lane) 11
**else**
12**if***i* is in the cut-in scenario13
$F_{ki}=\frac{E}{\Delta x_{i}}\cdot w_{k}=\frac{\frac{1}{2}w_{k}m_{i}v_{i}^{2}}{\Delta x_{i}}$
14Calculate $F=F+F_{ki}$ (the fore range only considers the same lane and the adjacent lanes)15
**end if**
16
**end for**
17Define the threshold force of the warning $F_{warning}$ and braking $F_{braking}$ based on the existed algorithm [17]18**if**$F(x_{i}$, $y_{i})-F_{ki}(x_{i}$, $y_{i})>F_{warning}$**then**19Output (“Danger from the rear vehicle,” warning in vehicle i) 20Output (“Attention the front vehicle,” warning in vehicle i−1)21
**else**
22**if**$F(x_{i}$, $y_{i})-F_{ki}(x_{i}$, $y_{i})>F_{braking}$**then**23Output (“Danger from the rear vehicle,” warning in vehicle i)24Output (“Attention the front vehicle,” warning in vehicle i−1 & active braking)25
**end if**


## 4. Experiment and Result Analysis

### 4.1. Experiment Setup

The experimental equipment includes three sedans as shown in Figure 5, namely a Honda Accord sedan (vehicle i) and two Changan Yuexiang sedans (vehicle j and k). Vehicle j and k were equipped with a GPS antenna on the roof of the vehicle body, and the GPS receiver, wireless safety unit (WSU), and an industry computer in the trunk. The industry computer is used for data collection and recording such as GPS position, CAN data, and time readings. The WSU is used for data transmission and sharing between vehicles. All the equipment of vehicle i are installed in its trunk. The overall architecture of the V2V communication system is shown in Figure 6, and all of the vehicle information is aggregated to WSU by cable. We choose DSRC devices as wireless safety units in each vehicle platform. According to the detail about the definition of ACP and PCP in Section 2.1. If the force on one vehicle is greater than the threshold force $F_{warning}$, the industry computers in both ACP and PCP will send signals to the voice prompt modules to alert drivers that risk is coming. Furthermore, if the force on one vehicle is greater than the threshold force $F_{braking}$, the industry computers in both ACP and PCP will not only send signals to the voice prompt modules to alert drivers, but the ACP will also receive an active braking signal to avoid the traffic risk. 

### 4.2. The Experimental Process

The field experiment route was the ShuiNan road, ChangPing district, northwest of Beijing, we conducted two experiment scenario including car-following scenario and cut-in scenario. The real experiment scenario is shown in Figure 7.

(1) Experiment 1: Car-following scenario

As shown in Figure 8, the participant driver was required to drive vehicle i following vehicle j. According to the sine equation $v_{j}=45+20\sin\left(2\pi t/T\right)\;\mathrm{km}/h$, the speed of vehicle j is automatically controlled. Moreover, we set that the period T(s) belongs to [24,48], and changes as {(24, 36), (30, 30),(32, 28),(25, 35),(34,26),
(27,33)} after every period. For example, in the first period, the stochastic model chose 24 or 36 as the period T. At the end of this cycle, the stochastic model chose 30 as the period T in the second period. Then, in the third period, the stochastic model chose 32 or 28 as the period T, and so on. The CAN data of vehicles i and j were recorded in the process of experiment. Meanwhile, these two vehicles shared their driving data real time through WSU antennas in the process of experiment.

(2) Experiment 2: Cut-in scenario

The cut-in scenario is shown in Figure 9. The participant driver was required to drive vehicle i to follow vehicle j with an adjacent vehicle k driving nearby. Similarly, vehicle j moved according to the sine equation $v_{j}=45+20\sin\left(2\pi t/T\right)\;\mathrm{km}/h$ with its velocity controlled automatically. Meanwhile, vehicle k drove in the adjacent lane and made cut-in actions when there was enough space between vehicle i and vehicle j during the experiment. In addition, the content and equipment of data collection and transmission is consistent with experiment 1.

### 4.3. Experiment Result

To ensure the safety of the experimental processes, we only activated the warning function of the V2V communication system; The driver makes the brake motion under voice prompt. The time-varying traffic risk map in the car-following scenario is shown in Figure 10, the traffic risk caused by the following vehicle i and the leading vehicle j are illustrated explicitly in the risk map. In this experiment, the system alerts the subject driver to focus on the vehicle ahead by the voice prompt modules when the equivalent force is over the threshold force of the warning $F_{warning}$. 

The relationship between the equivalent force and the traffic risk is studied by analyzing the motion state of the vehicle. In this paper, we randomly selected a segment of 200 s driving data. In this experiment in a car following scenario, the speed of vehicle j was automatically controlled according to a sinusoidal equation. As shown in Figure 11a, as the speed increases, so does the equivalent force, and vice versa. That is because of the equivalent force is proportional to the speed according to the defined expression logic. Figure 11b shows the relationship between the time headway and the equivalent force $F_{i}$ at the position of vehicle j. The curves illustrate that the equivalent force $F_{i}$ is inversely proportional to the time headway. As we know, the time headway is a classic risk assessment method. The larger the value of THW is, the safer the driving process is. The experimental results show that the method of traffic risk assessment based on the algorithm of equivalent force has the same effectiveness as THW in the traffic risk assessment in the car-following scenario.

The time-varying traffic risk map of an instantaneous cut-in scenario is shown in Figure 12, the traffic risk caused by the following vehicle i, the leading vehicle j, and the adjacent vehicle k are illustrated explicitly in the risk map. In this experiment, the system alerts the subject driver to focus on the vehicle ahead or adjacent by the voice prompt modules when the equivalent force is over the threshold force of the warning $F_{warning}$. The velocity of the leading vehicle is subject to the same equation in the car following scenario. The only difference with the car following scenario is that there will be a vehicle in the adjacent lane, and this adjacent vehicle k will cut in front of the vehicle i if a sizable gap between vehicle i and vehicle j becomes available. 

The relationship between time headway and equivalent force $F_{i}$ at the position of vehicle j in the cut-in scenario is shown in Figure 13. As we know, the leading vehicle is moving periodically, and the cut-in action of the adjacent car is also periodic. The experiment results illustrate that the equivalent force still shows a cyclical phenomenon. However, the curve of time headway has no obvious regularity with time. It shows that there are limitations in using THW for traffic risk assessment in some complex scenarios.

### 4.4. Discussion

The real road traffic environment has many kinds of participants such as pedestrians, cyclists, and vehicles, which makes it an extremely complicated system. In this study, we only predicted the motion of vehicles. The further research should take the behavior and motion state of pedestrians and cyclists into account to make the road traffic assessment more accurate. Furthermore, the HMM algorithm can be replaced by any other machine learning prediction algorithms in this framework, such as dynamic Bayes, Markov Monte Carlo, and can even cooperate with trajectory prediction algorithms. Future research will focus on the comparison to find a prediction algorithm which is simpler and more effective for the framework. Generally, each vehicle has its geometrical parameters. However, we considered the vehicle as a mass point; to some extent, it will affect the accuracy of road risk assessment. Future study will take the geometry of vehicles into account. In addition, we only considered the rolling resistance and the air resistance in the described mathematical model, while slope resistance and acceleration resistance are also worth considering under some specific conditions.

## 5. Conclusion and Future Work

This paper presents a model for quantitative analysis of the road traffic risk, based on the equivalent force and artificial potential field. The focus is on real-time and precise assessment of the road traffic risk. The risk distribution of road traffic established by this model is more consistent with human’s subjective feeling because of its certain risk range. The road traffic risk has a specific distribution which could explain why a person does not feel dangerous when a vehicle passes slowly and steadily in front of her/him, but she/he does when the vehicle drives fast and abnormally. This phenomenon reflects the accuracy of the model. Meanwhile, this study proposed a novel framework for road traffic risk assessment based on the HMM prediction method, which can assess the road traffic risk in advance to some extent. The time-varying risk map can be applied to traffic risk prediction and assessment for the decision-making of the traffic management department and intelligent technologies to prevent road traffic accidents. 

Future work will mainly cover the following two aspects: One for modeling and calculating of the equivalent force between the vehicle and the road boundaries such as lane lines and barriers, and the other for experiment design and verify. Moreover, the experiments will focus on how to deal with the situation of alarm and braking failure caused by algorithm failure.

## Figures and Tables

**Figure 1 sensors-18-04313-f001:**
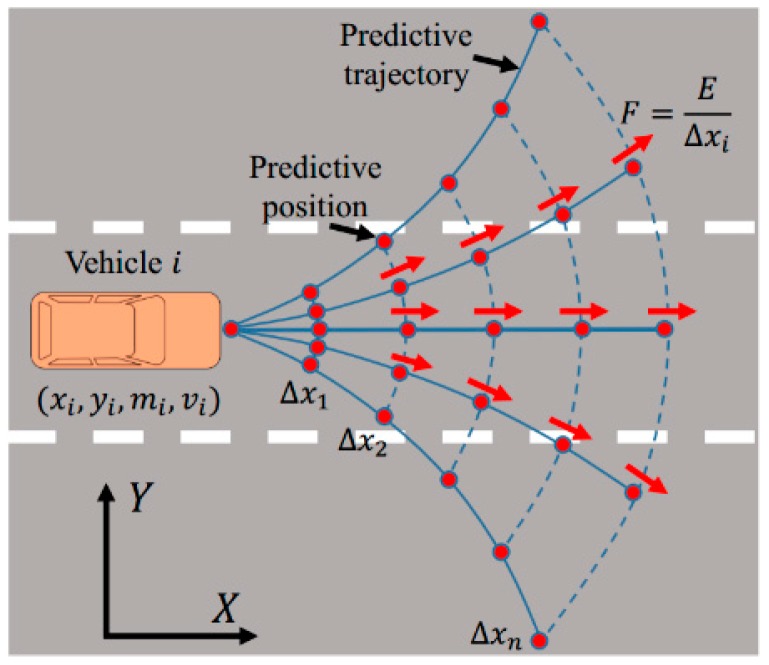
Predictive trajectories and positions diagram. The blue curve denotes the predictive trajectory, and the red point means the predictive position. Moreover, the predicted trajectories and points are generated under the assumption that the steering angle is constant. The red arrow indicates the equivalent force at each position. Δx is not only indicates the linear distance between vehicle i and others in a longitudinal direction but also denotes the arc length as shown in this figure.

**Figure 2 sensors-18-04313-f002:**
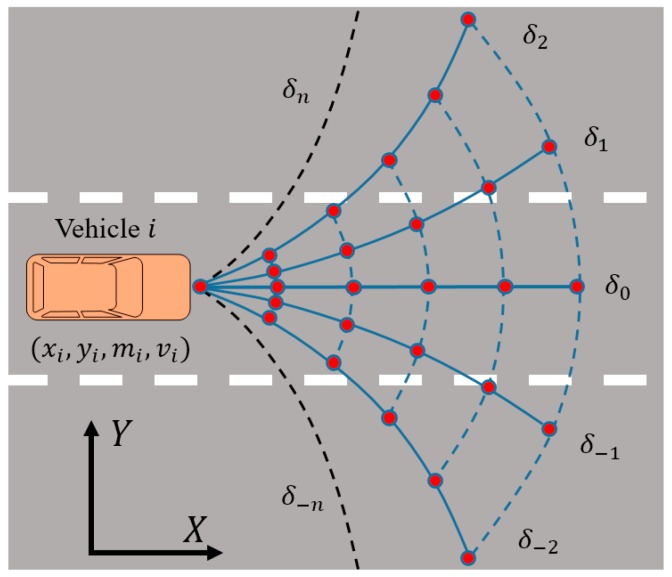
Trajectories based on the steering angle. $\delta_{i}$ denotes the steering angle corresponding to each predictive trajectory. The black dashed line is the predictive trajectory corresponding to the left and right limit steering angle, which is related to the vehicle speed and road conditions.

**Figure 3 sensors-18-04313-f003:**
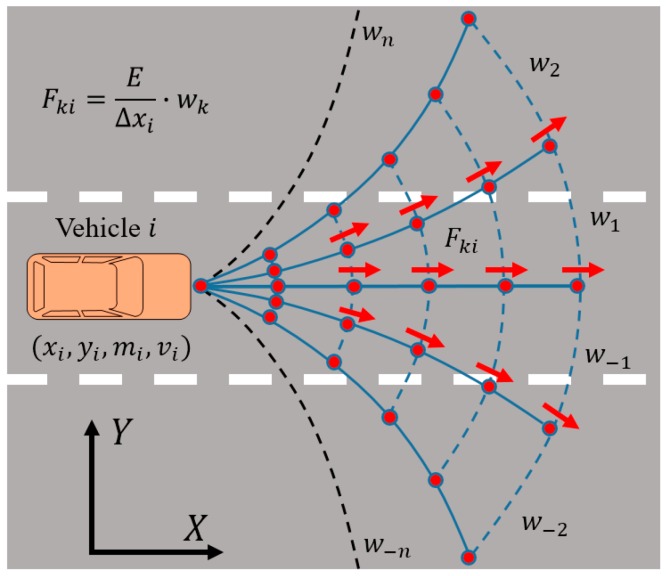
The setting of weights according to the HMM algorithm. It’s going to be challenging to solve for $S_{t}$ if we set the value of n too large. However, to make the calculation of equivalent force more accurate, we need to set the increment of steering angle more closely. So as long as the corresponding steering angle in the corresponding interval is $\left(\delta_{k-1},\delta_{k}\right]$ or $\left[\delta_{-k},\delta_{-k+1}\right)$, the weight will take $w_{k}$ or $w_{-k}$.

**Figure 4 sensors-18-04313-f004:**
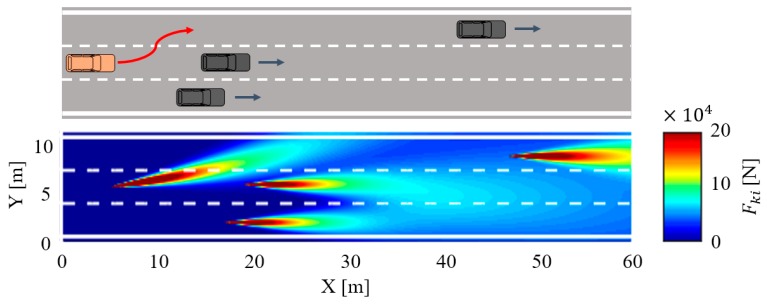
The time-varying traffic risk map. The three black vehicles are driving straight along the road lane. The orange vehicle follows the black vehicle in the second lane and is changing to the left lane.

**Figure 5 sensors-18-04313-f005:**
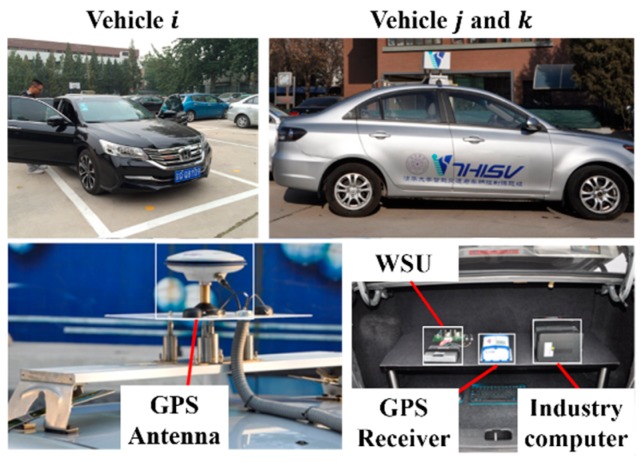
Experimental vehicles. WSU, GPS Receiver, and industry computer are installed in the trunk of each test vehicle.

**Figure 6 sensors-18-04313-f006:**
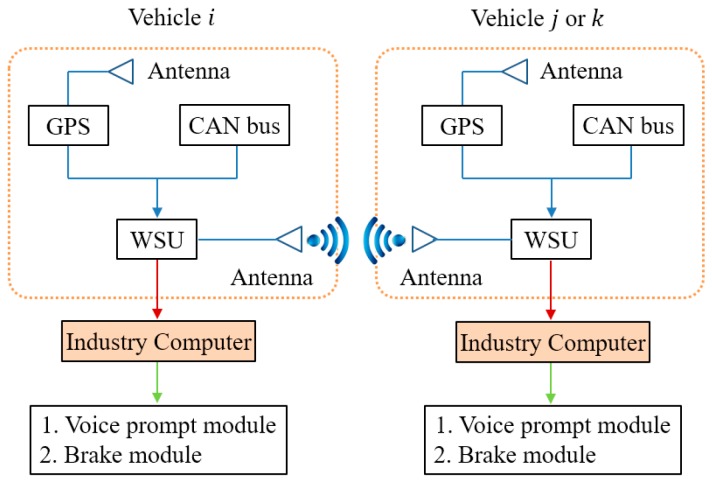
The overall architecture of the V2V communication system. The onboard equipment includes onboard GPS, CAN bus, and onboard WSU. All the driving information of the vehicle is collected to the vehicle WSU by cable, and experiment vehicles can share information through WSU by wireless transmission.

**Figure 7 sensors-18-04313-f007:**
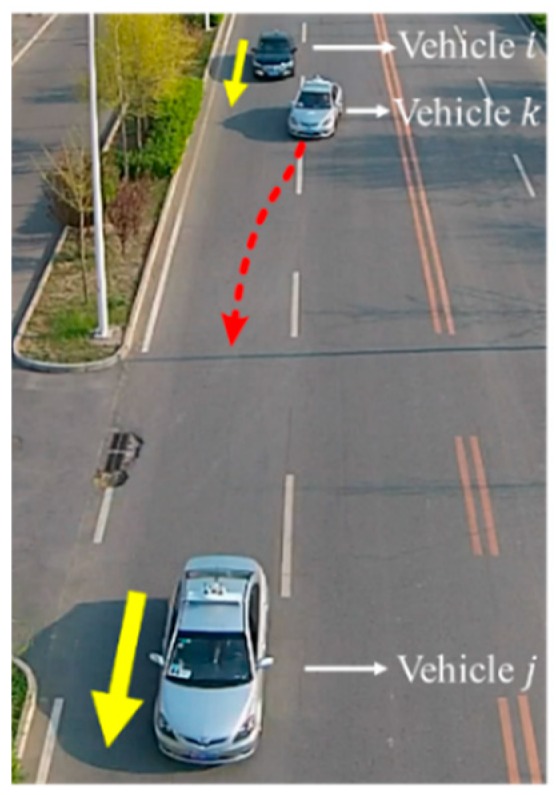
Actual scenario. The experiment site is a flat road on the outskirts of Beijing, which is in good road condition and surrounded by no other vehicles.

**Figure 8 sensors-18-04313-f008:**
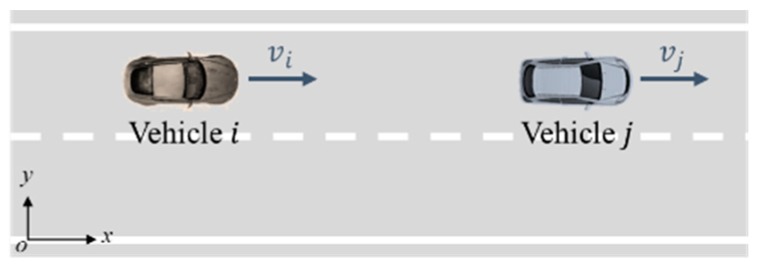
Car-following experimental scenario.

**Figure 9 sensors-18-04313-f009:**
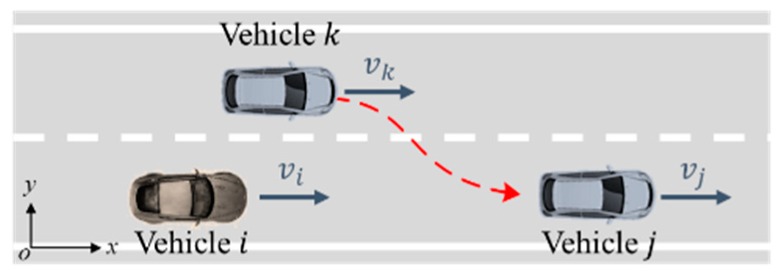
Cut-in experimental scenario.

**Figure 10 sensors-18-04313-f010:**
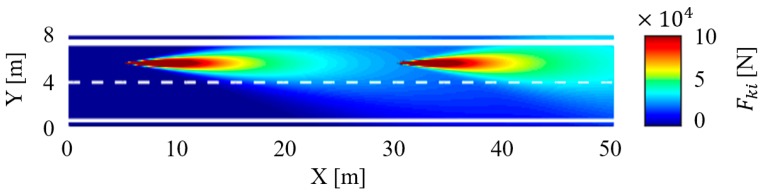
The time-varying traffic risk map in the car-following scenario.

**Figure 11 sensors-18-04313-f011:**
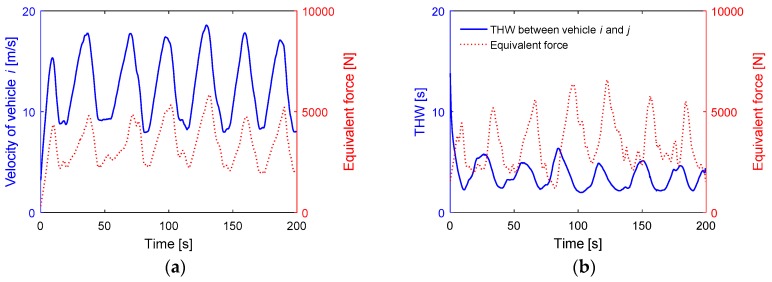
Experiment results in the car following scenario. (**a**) The relationship between the velocity of vehicle i and the equivalent force $F_{i}$ at the position of vehicle j; (**b**) the relationship between the time headway and the equivalent force $F_{i}$ at the position of vehicle j.

**Figure 12 sensors-18-04313-f012:**
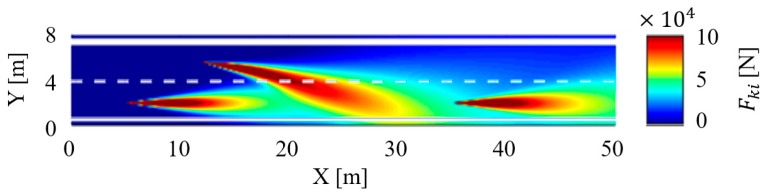
The time-varying traffic risk map in the cut-in scenario.

**Figure 13 sensors-18-04313-f013:**
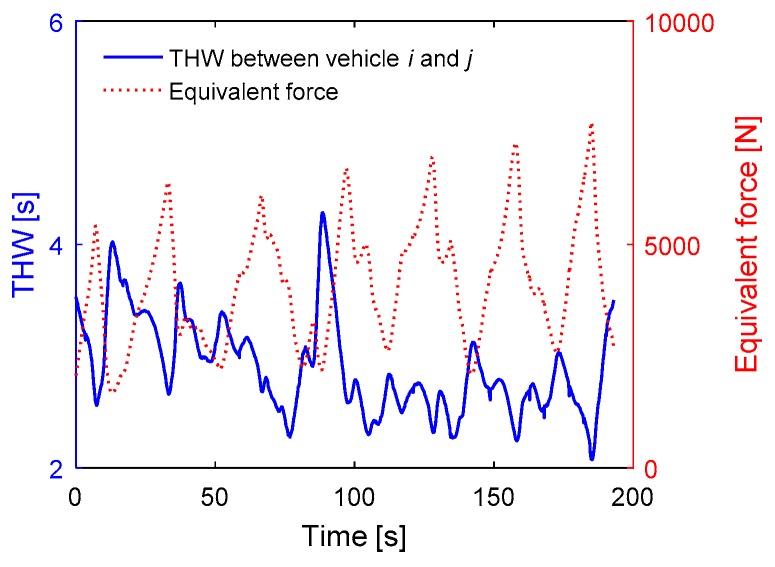
Experiment result in the cut-in scenario.

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
