# Peer review of "A Novel Framework for Road Traffic Risk Assessment with HMM-Based Prediction Model"

_sensors, 2018, doi:10.3390/s18124313_

Round 1
Reviewer 1 Report
In this manuscript, a road traffic assessment method is developed based on the Hidden Markov Model to predict drivers turning intentions according to the prevailing traffic conditions. The proposed method is intended to compute the road traffic risk for prevention of road traffic accidents.
In general, the article is clearly explained and well presented, however, the following concerns must be addressed before considering it for publication.
1. The contribution of the manuscript is rather weak. The probabilistic approach followed in the article is based on the dynamic windows method (See "The Dynamic Window Approach to Collision Avoidance") and resembles a recent article describing an obstacle avoidance assessment algorithm (See https://doi.org/10.1016/j.robot.2018.08.005). As such the authors should include these references and clearly state the contribution of their work.
2. The authors claim "we define the range of the potential traffic risk caused by vehicles according to vehicle kinematics which can describe the traffic risk consistent with human’s intuition perception of risk". To which extent this assertion has been validated? The author should include the necessary references to support such a statement.
3. In the definition of the forces and energy functions, the authors used the longitudinal position of the vehicles only ( the x-axis of the coordinate system). For the car-following case, this assumption is valid. However, it is unclear to me how this approach is applied during the cut-in maneuver.
4. Please explain how is Eq. (18) applied in the context illustrated in Fig. 3
5. How is the time-headway calculated?
6. In the stochastic model, how is the road information incorporated? Such an information is necessary to model the driver turning intention more accurately.
7. In the algorithm, please clarify further the forces used in steps 7, 9, and 10.
8. In lines 270-271, the period values are rather confusing.
Author Response
Dear reviewer:
We have answered one by one in response to your comments. Please refer to the attachment for details.
We would like to acknowledge that the improvement of this manuscript in terms of presentation clarity and quality is partially due to the constructive comments from you.
Thank you!
Yours sincerely
Authors

Reviewer 2 Report
The research sounds interesting since V2X is a trend topic in the research community. I have no particular issues about your paper, the topic is interesting, the case study scenarios, although simple, are adequate in order to give first insights on the methodology which is extensively described.
The only concern is about the description of results since you summarized all your work in just 2 figures. It is poor in my opinion since it is not helping the readers to understand if, and when, the algorithm detected the condition for Fwarning and Fbraking.
Describing with major details the results of your experiment will give more strength to your article.
Author Response
We strongly agree with the reviewer's opinion, and we appreciate your appreciation. We obtained a series of similar images to verify that the equivalent force method is more effective than THW algorithm. However, if we put too many similar figures in this manuscript may cause redundancy. Hence we only used a few pictures to illustrate the experiment results. Meanwhile, the purpose of this manuscript is to propose a framework for traffic risk assessment based on an HMM prediction model. We use field experiments to verify this framework is operationally feasible. To ensure the rigor of the study, we cannot determine the threshold in the current research stage exactly, because, to ensure the safety of the experiment process, there is no dangerous situation in our experiment. However, the scheme for determining the threshold value of F_warning and F_braking is referred to our previous research in the current research stage [1]. In this study, we just compared the effectiveness between equivalent force and THW algorithm. In the further study, we will focus on the threshold determination for F_warning and F_braking, and we will also pay our attention to the field experiment design and verify. Moreover, the experiment will also focus on how to deal with the situation of alarm and braking failure caused by algorithm failure.
[1] Wang J, Wu J, Zheng X, et al. Driving safety field theory modeling and its application in pre-collision warning system[J]. Transportation research part C: emerging technologies, 2016, 72: 306-324.

Reviewer 3 Report
It is a very good appraisal of a methodology that could be used to the defined purpose after a better inclusion of the reality, as it was recognised by the authors. Congratulations.
The main objective was start to develop an algorithm to predict road traffic conflicts/accidents and quantify the risk within “intelligent” vehicles, using a machine learning technique. At this point it is mainly interesting and to be relevant it needs, like the authors underlined, the consideration of other conditions, namely the involving of vulnerable users. The originality is given mainly by the machine learning technique used and the considerations made that were fairly proved by the tests.
Author Response
Thank you very much for your appreciation. We will also continue to improve the model and design new simulator and field experiments to improve the adaptability and effectiveness of the algorithm in various scenarios.
